# Drake-like Calculations for the Frequency of Life in the Universe

**Karl-Florian Platt**

Department of Media and Social Science, Fresenius University of Applied Sciences, 10117 Berlin, Germany; karl-florian.platt@ext.hs-fresenius.de

**Abstract:** This article is intended to provide a review of some modifications to the Drake equation, a 1961 concept presented by Frank Drake to determine the number of extra-terrestrial civilizations able to communicate. A reduced version of the Drake equation will then be presented. This can be used to estimate an important question for astrobiology, the frequency with which planets are habitable. Finally, a concept is presented that can also integrate the habitability of moons into the reduced Drake equation.

**Keywords:** Drake- equation; life; spectral class

## 1. Historical Background

In 1959, astronomers Giuseppe Cocconi and Philip Morrisson came up with the idea that the current measuring instruments could be used to receive signals from civilizations that have formed outside our solar system. These signals should be transmitted with the wavelength of neutral hydrogen 1.42 MHz due to the universal importance of this element [1]. More precisely, this is the 21 cm hyperfine hydrogen line. Obviously, a prerequisite for this is the existence of such civilizations and their ability to transmit light and/or radio signals into space. Two months later, astronomer Frank Drake and the National Radio Astronomy Observatory, Green Bank, now known as the Green Bank Observatory, tried to detect such signals. For this, he studied the stars Epsilon Eridani and Tau Ceti in the wavelength spectrum 1.42 MHz, as suggested by G. Cocconi and P. Morrison [2]. However, Drake was unable to detect any such signals. For his conference at the Green Bank Facility in 1961, however, he developed a conception, essentially the conference agenda, that would lead to an estimation of the appropriate kind. This conception was further discussed and elaborated in the following years by Drake and the nine other conference participants. The result of the work of this group, which often called itself the "Order the Dolphin", is the so-called Drake equation:

$$N = R^* \times f_p \times n_e \times f_l \times f_i \times f_c \times L \tag{1}$$

This can be used to calculate how many extra-terrestrial civilizations are able to communicate at a given time. The variable $R^*$ represents the star formation rate. This indicates the speed at which stars form from the material required for this purpose. The unknown $f_p$ is the proportion of stars in the universe around which planets form. The value $n_e$ indicates how many of these planets are in the habitable zone. Thus, the planets should have an appropriate distance to their star such that the stellar radiation (signified by the temperature in the first place) permits the existence of life. The variable $f_l$ then indicates the fraction of planets in the habitable zone where life actually arises, and $f_i$ refers to the proportion of such life forms that can be described as intelligent. Finally, $f_c$ is the relative frequency with which an intelligent life form is technologically advanced in order to be able to communicate with electromagnetic means. In addition, $L$ is the time length by

which such a sophisticated life form can send signals into space. The Order of the Dolphin thus calculated $R^*, f_p, n_e, f_l, f_i, f_e$ with the following values:

$R^* = 1$ Star per year.

$f_p \in [0.2; 0.5]$ (Between one-fifth and half of all stars have planets).

$n_e \in \{1; 2; 3; 4; 5\}$ (Stars with planets have between one and five habitable planets).

$f_l = 1$ (Life develops on all habitable planets).

$f_i = 1$ (Everywhere life is formed, intelligent life develops).

$f_c \in [0.1; 0.2]$ (Between 10% and 20% of all intelligent life forms communicate in such a way that measurable signals are released into space).

$L \in [1000; 100000000]$ (Intelligent life forms exist for between 1000 and 100 million years).

Taking the lowest values of each factor, the Order of the Dolphin received a minimum value of $N = 20$. If one takes the maximum value of each factor a maximum of N = 50,000,000 is received. That means there should be at least 20 civilizations in the Milky Way that send detectable signals into space [3].

## 2. Criticism of the Drake Equation

Over time, various arguments against the Drake equation have emerged. The most famous argument is named after the ItalianAmerican physicist Enrico Fermi. The so-called Fermi- paradox points to the serious difference between the number of detectable signals of extra-terrestrial civilizations estimated and the number of such signals actually detected so far [4]. This enormous difference suggests that the concrete values of the variables of the Drake equation were wrongly selected somehow. The Fermi paradox is therefore not to be understood as a critique of the method, but as a critique of the choice of variable values. If one wants to criticize the method itself, one should point out the extremely low data available to determine some variables used in the Drake equation. There are virtually no data on the variables $f_l$, $f_i$, and $f_c$ from which a realistic numerical value could be derived. To determine the value $f_l$ of habitable planets on which life is formed, one would have to observe a very large number of habitable planets over a long period of time. Another point of criticism is that the Drake equation ignores the fact that life could also form on the moons of planets. It cannot be ruled out that moons can be habitable as well. This could be the case even if the planet to which the exomoon belongs is not in the habitable zone of its star. We will discuss this in more detail in a later section.

## 3. Previous Modifications to the Drake Equation

Over time, several researchers have proposed modifications that can extend the Drake equation to include other aspects. Thus, in 1979, based on an idea by M. Hard, the chemists Clifford Walters, Raymond Hoover, and R. Kotra of the University of Maryland proposed to extend the Drake equation to include a component that can be used to map the will of a civilization to form colonies on other planets [5]. In 1983, Glen David Brin of the California Space Institute published a new version of the Drake equation [6]. He criticized the fact that the Order of the Dolphin does not take into account those problems that may arise when parallel evolving civilizations contact each other, nor that advanced life forms could spread around their home planet. His Drake equation takes this aspect into account. In May 2005, Alexander Zaitsev of the Russian Academy of Science proposed to add another factor $f_m$ to the Drake equation [7]. This value, known as the METI factor, is the proportion of extra-terrestrial civilizations capable of sending signals into space and consciously deciding to do so. In 2021, James Benford published a version of the Drake equation to calculate the number of alien artifacts instead of civilizations [8].

## 4. A Modified Drake Equation as Motivation to Pursue Astrobiology

As we have seen in Section 3, many modifications are moving in the direction of expanding the Drake equation with new components. At this point one should ask if it would not be appropriate in the sense of astrobiology to estimate the number of life-

carrying planets only, not referring then to the number of communicating civilizations. The latter seems to be an inappropriately high objective at the moment, which cannot be achieved with the methods and instruments currently available. If, on the other hand, one performs a modification of the Drake equation, so that only the parameters $R^*, f_p, n_e$, and $L$ play a role, one obtains a version that reflects the goals of astrobiology more adequately and consists for the most part of variables that can be determined realistically:

$$N = R^* \times f_p \times n_e \times f_l \times L \tag{2}$$

Here, $N$ stands for the number of planets in the Milky Way that can carry life rather than being the home for intelligent and communicating civilizations. In addition, $L$ is to be understood here as the lifetime of a planet and no longer as the length of the period during which extra-terrestrial civilizations send signals into space. As will be shown in the following sections, there are already usable data for these parameters. The only exception is the parameter $f_l$. There are almost no usable data for this one. However, it is precisely this fact that makes the Drake equation an important result of astrobiology, since the parameter $f_l$ must be seen as the quintessence of that kind of research. Unfortunately, the idea of reducing the Drake equation to fewer components runs counter to the effort to calculate the probability distribution of $N$ using the central limit theorem (CLT). This idea was raised in 2010 by Claudio Maccone and is the first attempt to investigate the Drake equation with the help of statistical theory [9]. As Manasvi Lingam and Abraham Loeb have argued, it is much more likely that we will detect various forms of life in general than intelligent life proper [10].

## 5. Data Situation of $R^*$

In this section, it will now be investigated whether certain statements can already be made about the frequency with which new stars form. In their summary of the research results so far, Robert Kennicutt Junior and Niel Evans II cite more than 300 technical articles on this topic [11]. It is therefore fair to say that the star formation rate is an intensively researched area. Based on the findings of these articles and other methods such as counting young stars, researchers come to the conclusion that a star formation rate of $R^* \in [1.5; 2.3]$ fits with the observations of the Spitzer telescope. It should be noted, however, that the star formation rate could also be two to three times higher, because bright, massive stars outshine the smaller, darker ones. The frequencies with which different star types form seem to play a decisive role. These frequencies are described by the so-called Salpeter mass functions (initial mass function). After these, the frequency of stars decreases exponentially with the mass of their types. For stars with more than half the mass of the sun, an exponent is assumed to be $\alpha = 2.3$ [12].

$$(M)\Delta M = \left(\frac{M}{M_S}\right)^{-\alpha} \tag{3}$$

where $M_S$ the mass of the sun. For stars with less than 0.08 solar masses, $\alpha = 0.3$ is assumed. If a star has between 0.08 and 0.5 solar masses, $\alpha = 1.3$ is used. In the Drake equation, therefore, one should not generally speak of one star formation rate. At this point, we would like to thank the referees for mentioning that there are other concepts to estimate R*. Instead of the Salpeter mass function, those proposed by Miller-Scalo [13], Kroupa [14], or Chabrier [15] could be used. Until now, it has been an open field of discussion as to which approach is the best. These different approaches may lead to different Drake equations as well. One also should discuss if one Drake equation for different star types is actually necessary. If so, a specially reduced version (as suggested in Section 4) would be necessary as well. In other words, maybe one should use a specific (reduced) Drake equation for each specific star type. The mentioned distinction in star types could also affect other components of the Drake equation or its reduced version. One referee noted that the stellar type could be included in the factor $f_l$ instead. That factor might tend to zero for the star

types O, B, and A. It also may tend to zero for stars of type M (the red dwarfs). We want to thank the referee for indicating this.

## 6. The Data Situation of $f_p$ and $n_e$

In 2012, a network of 42 scientists from renowned research institutes, including N. Kains of the European Southern Observatory Headquarters in Garching and S. Kane of NASA's Exoplanet Science Institute, Caltech, published a summary of the detections of extrasolar planets that had succeeded so far [16]. In addition, a research group around T. Sumi used a method called Gravitational Lens Effect in 2011 to detect at least as many Jupiter-type planets as there are known main sequence stars in the Milky Way [17]. From this, it can be concluded that it is the norm for stars to have planets. Following the NASA website for exoplanets, until now more than 4000 Exoplanets have been detected [18]. NASA's Kepler mission produced data from which Erik Petigura, Andrew Howard, and Geoffrey Marcy could derive the existence of 603 extrasolar planets. Ten of these planets, as the three scientists write in their technical article, are comparable to Earth in both their mass and the radiation dose they receive from their star [19]. The researchers also extrapolate that between 3.5% and 7.4% of all sun-like stars are orbited by Earth-like planets that take between 200 and 400 days to orbit. The researchers also estimate from the data that the product $n_e \times f_p$ should be close to 0.4. Using planetary statistics and the data from the Kepler mission, Wei Zhu from the Tsinghua University of Beijing and Subo Dong from the Peking University calculated that in the inner region of one AU (astronomical unit) or less, 30% of the Sun-like stars have planets with masses and radii down to Earth-size. They also calculated that planet systems have three of those planets on average [20]. From that point of view, the value 0.4 for $n_e \times f_p$ seems to be realistic if not arranged in a manner that is still too low. Lisa Kaltenegger from Cornell University mentioned that humanity now has the technology to detect exo-planets and will in the near future be able to detect habitable planets as well [21]. That is why one might be forced to recalculate the value of $n_e \times f_p$ in the near future.

## 7. The Data Situation of $L$

Let us first clarify what $L$ should describe exactly. Our goal is to find out how many planets there are that provide the conditions for the existence of life. Therefore, we would have to use $L$ to denote the length of time a planet is habitable. Since we do not yet know the exact conditions for habitability, we use the amount of time a star spends on the main sequence of the Russel–Hertzsprung diagram to estimate $L$. This makes sense if one assumes that planets form together with their stars and that habitable zones hardly change over the period $L$.

This period depends on the amount of hydrogen that the star must fuse into helium. This quantity is reflected in two components. For one thing, the more hydrogen it carries, the more massive a star is. On the other hand, the more radiation it emits through its fusion, the more luminous it is. However, mass and luminosity ($Lu$) are related by the following physical law:

$$Lu \sim M^{7/2} \tag{4}$$

Accordingly, the time $\tau$ a star spends on the main row can be represented as a function of the mass of a star.

$$L \approx \tau \sim 10^{10} years \times \left( \frac{M_S}{M} \right)^{5/2} \tag{5}$$

in which $M_S$ is the mass of the sun. This model was created by Michael Richmond and published on his own website [22]. At this point, we should keep in mind that this model predicts a lifetime of ten billion years for stars with one solar mass. The life on earth is known to be probably 4 billion years old. One should multiply the formula given above with a factor "a" to make the model more realistic. If we assume that life on earth is indeed 4 billion years old and that the sun existed for half of its total lifetime, a value of a = 0.6 should do the work. One could start a more detailed discussion at this point to find a

better value for "a". In 1983, Brandon Carter argued that intelligent life is exceptionally unlikely because there are "hard" evolutive steps for life to evolve intelligence [23]. The amount of planets on which intelligent life can develop depends on how many of these "hard" steps are needed. The lifetime of the star and of the biosphere its habitable planet is surrounded with has to be long enough, otherwise intelligent life cannot evolve. Based on that argument, Andrew Snyder-Beattie, Anders Sandberg, Eric Drexler, and Michael Bonsall developed a Bayesian model and with it calculated that most exoplanets should not have intelligent lifeforms [24]. Maybe models like these will lead to values of "a" that can make our Drake-like model even more realistic. The Snyder–Beattie model is also a good argument not to use Drake-like equations to calculate the number of communicating civilizations. As mentioned above, one should focus on general lifeforms.

## 8. First Evaluation and Extension to Moons

In Section 4, we proposed a reduced version of the Drake equation as Formula (2).

$$N = R^* \times f_p \times n_e \times f_l \times L$$

With the findings from the previous sections, we obtain the mass-dependent representation for sun-like stars. If such a star has the n-times the mass of the solar mass, we obtain

$$\frac{N(n \times M_S)}{pc^3} = n_e \times 0.6 \times 10^{10} \, x \left(\frac{1}{n}\right)^{\alpha+2.5} \times f_l \tag{6}$$

Let us use stars with three solar masses as an example. After the Salpeter function with $\alpha = 2.3$ from one cubic parsec gas over time,

$$R^* = (3)^{-2.3} = 0.0799 \tag{7}$$

Stars with three solar masses are formed. According to Erik Petigura, Andrew Howard, and Geoffrey Marcy, we can use $n_e = 0.4$ and thus get 0.03197 planets that are in the habitable zones of their stars. The entire time these stars spend on the main sequence in the sense of Hertzsprung–Russell is about 385 million years. During this time, life can develop around stars with three solar masses. For the number of habitable planets that are formed according to this model per cubic parsec, one obtains:

$$\frac{N(3M_S)}{pc^3} = 12303515.45 \times f_l \tag{8}$$

The reduced Drake model is now to be expanded so that the potential habitability of moons can also be taken into account. Two questions appear to be answered. For one thing, it is important to know how large the proportion of planets is that have moons around them. We will be describing this proportion as $f_M$. To do so, one would have to clarify what proportion of moons are habitable. A corresponding model would have the form:

$$N = R^* \times f_p \times \left(n_e \times f_{lp} + f_M \times n_M \times f_{lM}\right) \times L \tag{9}$$

where $f_{lp}$ is the proportion of habitable planets on which life actually forms, and $f_{lM}$ is the proportion of habitable moons on which life forms can be found. Currently, there are no celestial bodies outside our solar system known that are certainly moons. That is why no statistical statement about the value of $f_M$ can be made. Thus, theoretical considerations still need to be used to obtain an estimate of the proportion of planets around which moons form. Robin Canup and William Ward developed such a theory for gas giants [25]. The authors describe a process in which moons, when they are formed, are fed gas, and at the same time, moons are lost due to the orbital decay of the gas. Canup and Ward suggest that the moons of almost all gas giants thus have a total weight in the range of one-thousandth

of the planet's mass. Another formulation of this conclusion is that in the case of gas giants, a value $f_M$ is also close to 1.

The habitability of exomoons is a very young, almost unexplored area. Only a few articles have been published on this subject. Rory Barnes of the University of Washington joined Rene Heller in 2013 to introduce the term "circumplanetary habitable edge" [26]. This planet-surrounded habitable boundary describes how far a moon can be from its planet so that it still can be made habitable by it. The idea is that the planet heats its moon by tidal forces and reflects radiation from the star and emits heat radiation to its moon. From these theoretical considerations, it becomes clear that there must be a whole series of explicit conditions for a moon to be habitable. Therefore, $n_M$ is a rather small value or a random variable with rather low mean. One of the referees mentioned that missions to the moons Europa and Enceladus are in the planning stage. Observations of these two moons have shown that these two could be habitable. The lack of observations of moons outside our solar system, as mentioned above, means that it is not yet possible to make a statement as to how often or with what probability these explicit conditions are met. The model that we can form is

$$\frac{N(nM_S)}{pc^3} = \left( 0.4 \times f_{lp} + n_M \times f_{lM} \right) \times 0.6 \times 10^{10} years \times \left( \frac{1}{n} \right)^{\alpha+5/2} \tag{10}$$

It is hoped that in the future, observations and sufficiently well-founded theories will help to make the still open components more precisely identifiable. With the current state of knowledge, the model cannot be further clarified. Research in the field of astrobiology will help to determine the components $f_{lp}$ and $f_{lM}$. Research in the field of astrophysics will help to make the component $n_M$ calculable and will also allow us to create Drake equations for situations in which not only Earth-like celestial bodies are considered. Philosophical and socialscientific research could help to rediscover a model, as Drake and the Order of the Dolphin had in mind. Another interesting idea is to calculate the component of the Drake equation with statistical methods. Such calculations have been performed by Maccone [9] and by Xiang Cai, Jonathan Jiang, Kristen Fahy, and Yuk Yung [27]. A next step for our Drake-like model could be to calculate its component with those statistical techniques.

**Funding:** This research received no external funding.

**Conflicts of Interest:** The author declares no conflict of interest.

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
