# Peer review of "Drake-like Calculations for the Frequency of Life in the Universe"

_philosophies, doi:10.3390/philosophies6020049_

Round 1

Reviewer 1 Report

The author states that the Drake equation can be used to calculate how many signals of extra-terrestrial civilizations should reach Earth per unit time. This is incorrect. Drake assumed a steady state, R is a rate and L a time, so the equation gives the number of communicating societies in the galaxy.

His best idea is that different versions of the equation may be better used depending on the types of stars being considered.  Parts of the paper are poorly expressed. At one point he says "it is the norm for planets to have planets." Perhaps his native language is not English, but statements like this cannot be overlooked.

Author Response

Thank you for your comments. You are right if you suspect that English is not my mother tongue. I have fixed the two bugs you mentioned in the new version of my article. I have also tried to find better wording.

Reviewer 2 Report

I have read the manuscript by Platt several times so as to better understand the argumentsthat were made for modification of the Drake equation. The author discussed a number of interesting ways the Drake equation can be modified to address broader questions including the habitability of moons and a brief mention “that life could also form on the traitsof plants”. An interesting publication that discusses the search for non-intelligent life that might be of interest to the author and could be cited here: Lingam, M. and A. Loeb. 2019. Relative likelihood of success in the search for primitive versus intelligent extraterrestrial life. Astrobiology 19:28-39.This paper discusses alternate life forms that are not intelligent life and may also be relevant to some of the authors other discussions including life on moons and the Drake equation that is focused on the presence of intelligent life.The author also mentions METI, or extraterrestrial civilizations with the technology to send signals. There are more recent papers on this topic that the author could possibly acknowledge. Benford, J. 2021. The Drake equation for alien artifacts. Astrobiology 21(9)1-7.Socas-Navarro, H. et al., 2021. Concepts for future missions to search for technosignatures. Acta Astronautica 182:416-453.The author also highlights the component of the Drake equation thatindicates that life develops on all habitable planets as a parameter we know nothing about. I agreewith this since even our ideas about a habitable exoplanet is based on it being a rocky planet and at a distance from its sun to be able to retain water. Much emphasis by the astrobiology community studying exoplanets is on biosignatures that may be reflected in the atmosphere of these potentially habitable exoplanets much reliance on the James Web telescope to make these measurements.Another paper that might be of interest in that it proposes to use statistics to estimate the occurrence of extraterrestrial intelligence. I am interested in what the author thinks about using statistics in his discussion of ??, ??and ??, and particularly ??. See Cai, X. et al., 2021. A statistical estimation of the occurrence of extraterrestrial intelligence in the Milky Way galaxy. Galaxies 9,5 pages 1-14.I admit that I am a bit ambivalent about the use of statistics for these variables, but I am also aware that the astrobiology community will have to make decisions about what planets to put emphasis on in the search for life. I know that there is considerable interest in focusing the James Webb telescope when available on the Trappist planetsthat are in the habitable zone based on the potential to retain surface water.I am also curious what the author thinks about in the section discussing R* and the frequency of stars, and the difference in ageing of the sun in our solar system versus white dwarfs that could last for tens to hundreds of billion years. How would a very long-lasting white dwarf with planets in the habitable zone effect parameters in the Drake equation? I am pleased that the author brought up the possibilities that moons should be considered as possible habitable bodies. It may be useful to the reader to mention the moons in our solar system in which future missions to search for life are in the planning stage: Europa and Enceladus. These moons are important because if life exists, it would be in the subsurface and driven by water/rock reactions such as serpentinization that is maintained due to the flexing of the moons during orbit due that could result in the mimicking ofsubduction on Earth that is

Reviewer 3 Report

This paper presents a reduced form for the Drake equation. The original Drake equation is an attempt to quantify the number of civilizations able to communicate at interstellar distances that exist simultaneously in the Galaxy.
The author shifts the focus of the equation from technological civilizations to inhabited planets. He tries to give an estimate of such a number using available astronomical data. He then proposes to extend this equation to possibly habitable Moons.

The subject of the paper is interesting, if not particularly original. Unfortunately, the core result, the estimate of the value N of inhabited planets, is based on only one source of data on exoplanets. Moreover, such a source is outdated: it is a paper published in 2012. Nine years, in the ever growing field of exoplanets search and characterization, is a very long period of time.
The proposal of including exo-moons in (reduced) Drake  equation is, to my knowledge, original. For this very reason it would deserve a more detailed discussion.

Given these considerations, I cannot suggest this paper for pubblication in its current form. I think that a revision is needed, especially to use updated exoplanets properties and to expand the discussion on the inclusion of Moons into the equation.

Detailed list of issues:

- section 1, "wavelength of neutral hydrogen 1,42 Mhz" More precisely, this is the 21cm hyperfine hydrogen line (spin flip of the hydrogen's proton). Perhaps stating it would be clearer.

- section 1, equation: It would be better to state the meaning of every variable in the equation. Also, please NUMBER the equations in the paper.

- section 2: what is "the Trabant"?

- section 3: I suppose that "number of life forms" stands for "number of inhabited planet". Otherwise, it would refer to the number of living beings and this is clearly not the target here

- section 5: the author chooses one specific initial stellar mass function, the Salpeter one. But there are many different IMFs that have been proposed. From the theoretical point of view, this issue is currently unsolved.
From the observational one, one could quote Miller-Scalo(1979), Kroupa (2001), Chabrier (2003). Results may significantly differ if one among the other IMFs were chosen.

- section 5: "one should use a own Drake equation for different star type":
Not necessarily. The stellar type could be included in f_l : this factor shoud go to zero for stellar type O, B, A;  there is a strong discussion in literature on stellar type M (red dwarfs). 

- section 6: This is the main problem of the current formulation of this work. The field of exoplanets discovery and characterization is subject to a fast growth. From 2012 to now, things have changed very significantly. 
See eg the sites
http://www.exoplanets.org/
https://exoplanets.nasa.gov/
for a updated view of our knowledge on this subject.
Recent review papers are also available, see e.g. Zhu & Dong 2021; Meadows & Barnes 2018; Kaltenegger et al 2019; and references therein.

- section 7: the author introduces an arbitrary factor "a" to obtain the duration of habitability of a planet given its central star type.
Things can be made more qualitative here. Usually, habitability is connected with the presence of liquid water at the planetary surface. As far as the Earth is concerned, the first ~500 Myr are excluded from the habitable period because the planet is too geologically hot to allow for the presence of liquid water. 
Then, considerations are often made on the fact that a main sequence star changes (actually, increases) its luminosity with time. In particular the Sun is now ~ 25/30% more luminous that 4 Gyr ago.
Based on this fact, computations show that the Earth should  lose its water in about one billion year  from now, just because Sun's luminosity is increasing. Similar considerations can be applied to other stars and planetary systems.
See e.g. Snyder-Beattie et al 2021 and references therein for discussion on this point and related ones.

- section 8: I again suppose that "number of lives" here, stands for "number of inhabited planets"

- section 9: as stated above, a deeper discussion should be given for the last formula (please NUMBER them) that extend Drake's equation to exo-moons.

Round 2

Reviewer 1 Report

The presentation is much clearer, and the main objection I had has been addressed.

Author Response

Thank you again for helping me to find the right wording.

Reviewer 3 Report

The paper has been considerably improved. The author addressed my criticisms, improved bibliography and data sourced and clarified the test. I now suggest this paper for publication.

Only, a number of very minor things should be corrrected.

line 293, pag. 4: concepts -> those proposed by

line 388 pag. 5: row -> sequence

line 398 pag 5: "In 1983 Brandon-Carter... cannot evolve.". This is not completely exact. The idea is that there are a number of "hard" evolutive steps, and the amount of planets on which intelligent life can develop depends on how many "hard" steps are needed to evolve intelligence, how difficult those steps are, and how long is the lifetime of the planet's biosphere.

line 394 pag. 5: formular -> formula

line 466 pag. 6 -> main series -> main sequence

line 497 pag. 6: characteristic -> mean

Author Response

Thank you for the new list of corrections. I corrected these lines in the new version.